# Biodegradable and pH Sensitive Peptide Based Hydrogel as Controlled Release System for Antibacterial Wound Dressing Application

**DOI:** 10.3390/molecules23123383

**Published:** 2018-12-19

**Authors:** Jie Zhu, Hua Han, Ting-Ting Ye, Fa-Xue Li, Xue-Li Wang, Jian-Yong Yu, De-Qun Wu

**Affiliations:** 1Key Laboratory of Textile Science and Technology, Ministry of Education, College of Textiles, Donghua University, Songjiang District, Shanghai 201620, China; zj910205@126.com (J.Z); hanhua8983309@163.com (H.H.); dhuytt@sina.com (T.-T.Y.); 2Modern Textile Institute, Donghua University, Changning District, Shanghai 200051, China; wxl@dhu.edu.cn (X.-L.W.); yujy@dhu.edu.cn (J.-Y.Y.)

**Keywords:** hydrogel, peptide, acrylic acid, biodegradable, pH-sensitive

## Abstract

The stimuli-sensitive and biodegradable hydrogels are promising biomaterials as controlled drug delivery systems for diverse biomedical applications. In this study, we construct hybrid hydrogels combined with peptide-based bis-acrylate and acrylic acid (AAc). The peptide-based bis-acrylate/AAc hybrid hydrogel displays an interconnected and porous structure by scanning electron microscopy (SEM) observation and exhibits pH-dependent swelling property. The biodegradation of hybrid hydrogels was characterized by SEM and weight loss, and the results showed the hydrogels have a good enzymatic biodegradation property. The mechanical and cytotoxicity properties of the hydrogels were also tested. Besides, triclosan was preloaded during the hydrogel formation for drug release and antibacterial studies. In summary, the peptide-based bis-acrylate/AAc hydrogel with stimuli sensitivity and biodegradable property may be excellent candidates as drug delivery systems for antibacterial wound dressing application.

## 1. Introduction

Hydrogels with three-dimensional, interconnected and polymer networks have the advantages of high-water absorption capacity and biocompatibility, which can be investigated for diverse biomedical applications including tissue engineering, axonal regeneration, wound dressings and controlled drug release [1,2,3,4,5]. Among various applications, hydrogels with the crosslinked structure, similar to the native extracellular matrix (ECM), are promising candidates for wound dressings, because hydrogels can create a moist environment, absorb wound fluids and facilitate auto debridement of wounds by rehydrating slough and enhancing the rate of autolysis [3,6,7].

In recent years, the stimuli-sensitive hydrogels, which can regulate the drug release ratio by responding to environmental changes such as pH, temperature and ionic strength, have achieved great attention in controlled drug delivery [8,9,10,11]. Especially, the pH-sensitive hydrogels, of which swelling behavior depends on environmental pH, have been proved useful for targeting drug delivery systems by changing their swelling and deswelling behavior. In an acidic environment, the hydrogels with a lower swelling ratio remain collapsed and the drugs will remain protected, while with the increasing of pH, the hydrogels will swell at a faster drug release rate [12,13,14]. This swelling–deswelling performance is commonly utilized as target drug delivery carriers at desired sites [13,15,16]. For example, the chronic wounds and infected wounds with a high bacterial load show a slight alkaline pH value (>7.3), and therefore, the hydrogels will release drugs at a faster rate to kill the bacteria in a short period; in contrast, wounds with pus or necrotic tissue (such as ulcers) show an acidic pH value so that the hydrogels will release the drugs in a slower and more steady profile [17].

Compared with non-degradable hydrogels, the biodegradable hydrogels which can degrade into safe molecules are more coincident with the requirements of biomedical applications, as they can be introduced into the bodies with minimal invasion [18]. Besides, as a drug release system, the release rate of drugs from biodegradable hydrogels can be controlled by several factors such as the enzyme concentration and crosslink density [4,19,20]. Moreover, the hydrogel degradation can allow for hydrogel clearance after drug exhaustion [9,19]. Herein, we anticipate constructing a hydrogel combining pH sensitivity and biodegradation properties for controlled drug release and wound dressing application [21,22,23].

Up to date, the peptide-based hydrogels have the advantages of low cost, easy preparation and structural diversity which can be applied in biomedical applications [24,25,26,27]. Moreover, the peptide-based hydrogels are highly biocompatible; when they biodegrade, the degraded products are with no apparent toxicity [28]. Most importantly, the degradation rate of the peptide-based hydrogels can be carefully tuned by the monomer design and the concentrations of enzymes such as trypsin and α-chymotrypsin for catalyzing the hydrolysis of amide bonds [29,30]. Considering these effects, endowing the biodegradable peptide-based hydrogels with pH-responsive property can be designed for drug release and wound dressing application.

In this study, we developed a series of pH-sensitive hybrid hydrogels by incorporation of the peptide-based bis-acrylate as a crosslinker into the hydrogels, and the peptide-based bis-acrylate chains could be biodegraded by enzymes (Figure 1). The amino acid sequence GGGGGGGK with acrylate end-groups, used as a crosslinker, was first synthesized by solid-phase reaction. Then, the hybrid hydrogels with different feed ratios of the crosslinker and acrylic acid (AAc) were prepared. The chemical structure of the peptide-based bis-acrylate was investigated by hydrogen-1 nuclear magnetic resonance (^1^H NMR). The morphologies of the hybrid hydrogels were observed by scanning electron microscopy (SEM) and the swelling ratio in response to pH change was measured. The biodegradation, cytotoxicity (in vivo and in vitro) and drug release of the hydrogels were characterized. Finally, triclosan was preloaded in the hydrogels for antibacterial application.

## 2. Results and Discussion

### 2.1. Synthesis of the Peptide-Based Bis-Acrylate and Hydrogels

To prepare the pH-sensitive and biodegradable hydrogels, the peptide-based bis-acrylate was first synthesized as a crosslinker, as shown in Figure 2. The chemical structure of peptide GGGGGGGK and peptide-based bis-methacrylate was characterized by ^1^H NMR. As shown in Figure 3a, the peaks at 5.5–6.5 ppm indicated the acrylate of the peptide which demonstrated the successful double bond coupled to the end of the peptide. The other corresponded peaks were analyzed and given in Figure 3a. After successfully synthesizing the peptide-based bis-acrylate, the hydrogels (Gel-1, Gel-2, Gel-3 and Gel-4) were then prepared, and their feed ratios are listed in Table 1. To confirm the hydrogel formation, FTIR spectra were characterized as shown in Figure 3b. The results showed that the peaks (peaks a and b) at near 1650 and 1540 cm^–1^ could be seen for all hydrogels which were assigned to typical amides I and II [3,31]. That is because the peptide-based bis-methacrylate was formed via the amido bond.

In this study, a novel peptide-based bis-acrylate formed by multi-peptide bonds composed of several repeated glycine and lysine was prepared. Benefited from this design, the peptide-based bis-acrylate, as a crosslinker, has the advantages of biocompatibility and enzymatic biodegradation. Besides, AAc-based hydrogels have been proved to be pH-sensitive in previous studies because the ionizable acid groups in hydrogels can accept and donate protons in response to the change of pH [12]. The combination of peptide-based bis-acrylate and AAc in this work would endow the hydrogels with both pH-sensitive, eco-friendly and biocompatible abilities, which are highly attractive in controlled drug release and wound dressing applications.

### 2.2. Morphology Investigation and Swelling Ratios of Hydrogels

To evaluate the morphologies of the peptide-based bis-acrylate/AAc hydrogels before biodegradation, the swollen hydrogels were freeze-dried and observed by SEM. The freeze-dried hydrogels exhibited well-defined interconnected and porous structures, as shown in Figure 4, and the pore sizes changed accordingly with different feed ratios of peptide-based bis-acrylate and AAc. The pore sizes of Gel-1, Gel-2, Gel-3 and Gel-4 were about 9.78 ± 1.59, 7.22 ± 1.46, 4.79 ± 0.86 and 3.00 ± 0.46 μm, respectively (Appendix A). It could be seen that with the increasing of peptide-based bis-acrylate contents, the pore size of the hydrogels would decrease. That is because with the increasing of the crosslinker concentration, the distance between polymer chains would decrease, leading the pore size to decrease and crosslinking density to increase, which was consistent with the results in Appendix A calculated by Equation S1 and Equation S2. As the properties of hydrogels (such as the swelling ratio, mechanical property and in vitro release) are highly related to the pore size and crosslinking density of the hydrogels [6,32], this design of the peptide-based hydrogels in this work makes the hydrogel more controllable for various applications including drug release and wound dressings.

Water uptake capacity is one of the most essential properties of hydrogels for wound dressing application [33,34,35]. Adequate swelling property can create a moist environment which can increase the epithelialization rate of wound sites. In this study, as AAc-based hydrogels are pH-sensitive, the pH-dependent swelling behavior of the hydrogels was investigated. Seen from Figure 5, all the hydrogels exhibited a quick and sudden increase in swelling ratio in a short period. At a time period of 4 h, the swelling ratios of Gel-1, Gel-2, Gel-3 and Gel-4 were about 10.56, 8.68, 7.05 and 6.03, respectively, at pH = 3 (Figure 5a), about 15.15, 12.38, 10.78 and 8.44, respectively, at pH = 7 (Figure 5b), and about 19.22, 16.78, 14.89 and 12.76, respectively, at pH = 10 (Figure 5c). It could be found that for the same hydrogel, the swelling ratio would increase when the solution became alkaline. This phenomenon confirmed the pH sensitivity of hydrogels and the hydrogels had a better water absorption ability in the alkaline environment. In an acid environment, the AAc-based hydrogel network would become tighter and more compact because of the stronger hydrogen-bonding interactions among the free carboxylic acid groups of the hydrogels network, thus leading to a lower swelling ratio. In contrast, the hydrogen bonds would be broken, and electrostatic repulsion could repel chain segments in the hydrogel network in neutral or alkaline media, and therefore, the hydrogel would have a higher swelling ratio [12,36]. Moreover, the water uptake capacity would be reduced when increasing the feed ratio of the peptide-based bis-acrylate. That is because the hydrogels with more peptide-based bis-acrylate amounts had a denser structure caused by the higher crosslinking density that would weaken the water absorption ability. Benefited from the pH-responsive property, the hydrogels can be utilized as controlled release systems, and will have a better exudate absorbing ability when used at infected sites as wound dressings.

### 2.3. Enzymatic Biodegradation of Hydrogels

The enzymatic degradation behavior of peptide-based bis-acrylate/AAc hydrogel was investigated by weight loss and SEM observation in the presence of trypsin. As seen in Figure 6a, the peptide-based bis-acrylate/AAc hydrogels showed obvious weight loss with trypsin at a concentration of 0.1 mg/mL and the weight losses of Gel-1, Gel-2, Gel-3 and Gel-4 just after 24 h were 12.5%, 10.7%, 9.7% and 8.3%, respectively. After 8 days, Gel-1 almost biodegraded (weight loss of more than 90%) with the fastest degradation rate, compared with the other hydrogels. The weight loss amounts of hydrogels varied in order: Gel-1 > Gel-2 > Gel-3 > Gel-4. It could be seen that the biodegradation behavior of peptide-based bis-acrylate/AAc hydrogels mainly depended on the crosslinking density. The previous studies have proved the biodegradation property of hydrogels is influenced by the hydrogel network, that is to say, the numbers of crosslinkers, vinyl groups and biodegradable groups in the chain, and molecular weight of molecules all can affect the biodegradation behavior [32,37]. In this study, during the biodegradation of hydrogels, polymer chains would first be broken by enzyme cleavage; the broken segments would then gradually migrate out from the hydrogel networks, and dissolve in the solution finally, leading the weight of the remained network to decrease. As shown in Figure 6a, the degradation rate decreased with increasing peptide-based bis-acrylate amounts, i.e., higher crosslinking density. As a result, the hydrogel with a looser structure had a faster degradation rate in our study.

The morphologies of the hydrogels after enzymatic biodegradation were observed by SEM. The SEM images of freeze-dried peptide-based bis-acrylate/AAc hydrogels before biodegradation exhibited a highly macroporous structure in Figure 4. After enzymatic degradation, the crosslinker peptide-based bis-acrylate chains in the network would biodegrade. Compared to hydrogels before degradation, the pore size of the hydrogels after biodegradation increased largely and some of the walls were broken. Xu et al. prepared an injectable hydrogel based on two recombinant proteins (ULD-TIP1 and ULD-GGGWRESAI) and found this hydrogel could exhibit a relatively quick degradation rate even in the pure phosphate buffer saline (PBS) buffer [38]. Their studies showed the hydrogel was almost dissolved and/or degraded after 144 h in the pure PBS solution, which was too fast to meet requirements in the practical application. Compared with Xu’s work, the hydrogels in our work showed a more controllable and longer degradation process with a period of more than 8 days, which depended on the crosslinking density and enzyme concentration [6]. The biodegradation property of the hydrogels suggests the peptide-based bis-acrylate/AAc hydrogels are eco-friendly, which is suitable for drug release and wound dressings.

### 2.4. Meanchial Property and In Vitro Release

A good mechanical strength of hydrogels is critical in practical application; a weak and easily broken hydrogel will hinder their application as wound dressing. As the pH value in wounds can change with therapeutic interventions [17], the compressive property of the swollen and neutral peptide-based bis-acrylate/AAc hydrogels was tested. The compressive modulus data were calculated as shown in Figure 6a. The results showed Gel-1 had the lowest modulus and Gel-4 had the highest modulus data among the four hydrogels, which suggested that the mechanical property of hydrogels was highly related to their structure and the hydrogel with a higher crosslinking density and denser porous structure would have a better mechanical property. In this study, the mechanical of the peptide-based bis-acrylate/AAc hydrogels could be controlled by the feed ratio. As wound dressings, the prepared peptide-based bis-acrylate/AAc hydrogels have sufficient mechanical strength.

This pH sensitivity of hydrogels can be also utilized in controlled release. For this goal, the cumulative release was investigated by preloading triclosan into hydrogels during the formation of hydrogels. The drug entrapment efficiency (DEE) and drug loading efficiency (DLE) of Gel-1, Gel-2, Gel-3 and Gel-4 in a neutral environment were calculated as 44.28% and 0.61%, respectively, 47.31% and 0.63%, respectively, 50.14% and 0.68%, respectively, and 52.37% and 0.70%, respectively. The triclosan release profiles at 37 ^°^C in PBS solutions with various pH values were shown in Figure 7b–d. The results showed that all hydrogels, even in different conditions, exhibited a burst and sudden release in the first stage, especially in the first 24 h. In the following period, the release rate of hydrogels would become lower, and after 10 days, triclosan would be almost released completely. As shown in Figure 7b, the average cumulative release amounts of Gel-1, Gel-2, Gel-3 and Gel-4 were 68.49%, 62.79%, 53.33% and 45.64%, respectively, in an acidic solution (pH = 3). It could be seen that Gel-1 had the fastest release rate and Gel-4 exhibited the lowest release rate, and this trend could also be found in neutral and alkaline conditions (Figure 7c,d). As proved in previous studies, the release profile could be divided into two stages; the initial release is ascribed to the drugs near the hydrogel surface which is related to the pore size and the larger pores in diameter in the hydrogel are beneficial for drug to release immediately [6,39], resulting in the burst drug release; the stable release stage seems to be a diffusion process caused by the solution flowing and exchange. In this study, Gel-1 had a larger pore size and a looser structure, leading to the fastest release rate compared with other hydrogels. As the solution became neutral (pH = 7), the average cumulative release amounts of Gel-1, Gel-2, Gel-3 and Gel-4 were 70.48%, 65.55%, 56.57% and 49.58%, respectively. In addition, the cumulative release amounts would be higher when the solution was changed into the alkaline condition. It could be found that for the same category of hydrogel, the release rate of hydrogels in an alkaline condition was higher than that in an acidic condition. The release rate mainly depended on the structure of hydrogel, that is, the release ratio would increase with the increase of pore size and decrease of crosslinking density. Shang et al. have confirmed that the pore size of freeze-dried AAc-based hydrogel in alkaline buffer solutions would turn to be larger than that in acidic conditions [10]. The peptide-based bis-acrylate/AAc hydrogels in this study showed pH-dependent swelling property and the pore sizes would become bigger when immersed in the alkaline solution; therefore, the drug release rate would become higher compared with that in acidic conditions, which has potential applications in controlled release. When applied at the bacterial infected wounds (in slight alkaline conditions), the peptide-based bis-acrylate/AAc hydrogels will release drugs at a faster rate for killing the bacteria in a short period. When used at wounds with pus or necrotic tissue (in an acidic condition), the hydrogels will release the drugs in a slower profile, achieving a long-term drug release effect.

### 2.5. Cytotoxicity of Hydrogels and Cell Culture

The biocompatibility is one of the most important characteristics of hydrogels as biomedical materials. The cytotoxicity of peptide-based bis-acrylate/AAc hydrogels was evaluated by quantitative MTS assay, as shown in Figure 8a. The cell viability percentages of Gel-1, Gel-2, Gel-3 and Gel-4 were approximately 94.14%, 96.49%, 92.20% and 91.97%, respectively. The results exhibited that there was no significant difference (*p* > 0.05) between these four hydrogels and the cell viability of all hydrogels was higher than 90% after 24 h, indicating all the hydrogels had excellent biocompatibility. After 48 h, there was a slight decrease in cell viability for the hydrogels which still had no obvious toxicity. Besides, the cell viability of triclosan (T)-preloaded hydrogels (Gel-1-T, Gel-2-T, Gel-3-T and Gel-4-T) was also investigated as control. Seen from Figure 8b, the cell viability of triclosan-preloaded hydrogels all displayed a bit of decrease after 24 and 48 h. For Gel-4-T, the cell viability percentages after 24 and 48 h were about 85.92% and 79.03%, respectively, which did not generate severe cytotoxicity. In contrast, Gel-1-T showed 75.13% and 67.67% in cell viability after 24 and 48 h, respectively, which displayed the lowest biocompatibility among the four hydrogels. It was notably that the cell viability of Gel-1-T, Gel-2-T, Gel-3-T and Gel-4-T was related to the drug release profile; that is, the hydrogel with a higher release rate had a higher cytotoxicity caused by the released triclosan, which might cause damage to cells. However, the results also indicated the cytotoxicity of drug-preloaded hydrogels could be modulated by the feed ratio of peptide-based bis-acrylate and AAc. As a result, the good biocompatibility indicated the peptide-based bis-acrylate/AAc hydrogels and Gel-4-T in this study were good candidates for wound dressings.

Moreover, the L929 cells were cultured with Gel-1, Gel-2, Gel-3 and Gel-4 for 24 h, and Gel-1-T, Gel-2-T, Gel-3-T and Gel-4-T were also utilized as control. As shown in Figure 8c, the images showed that a large amount of cells were grown and proliferated after 24 h on the hydrogel surface. In contrast, for the control groups (Gel-1-T, Gel-2-T, Gel-3-T and Gel-4-T), the cell amounts had a decline after 24 h compared with the pure hydrogels, demonstrating the triclosan had slight cytotoxicity. There was also a little difference in cell amounts between Gel-1-T, Gel-2-T, Gel-3-T and Gel-4-T, which was in accordance with the results from the MTS assay as well as the drug release. Both of the MTS assay and cell culture indicated that the peptide-based bis-acrylate/AAc hydrogels have favor biocompatibility. Besides, the cell viability of drug-preloaded hydrogels could be affected by the crosslinking density of hydrogels; as the crosslinking density increased, the pore size of hydrogels would be decreased, leading the drug release rate to be slower.

Moreover, in vivo toxicity of Gel-1, Gel-2, Gel-3, Gel-4 and Gel-4-T was further examined after 24 h. The results in Figure 9 showed no significant inflammatory or toxicological responses; it could be observed that the key organs, including liver, spleen, kidney, heart and lung in all groups, displayed an integrated and complete tissue structure after 24 h, further confirming these peptide-based bis-acrylate/AAc hydrogels and Gel-4-T had no obvious toxicity in vivo which could be used safely as wound dressings.

### 2.6. Antibacterial Property of Hydrogels

Nowadays, it is better to endow the wound dressings with more functions based on the wound classification to accelerate the wound healing in practical application. Among these, antibacterial property is one of the most significant functions for treating the infected wounds. Studies have showed patients can be infected by microorganisms including *Pseudomonas aeruginosa*, *Staphylococcus aureus* and *Staphylococcus epidermidis* at the wound sites [40]. In this study, the antibacterial property of triclosan (as a drug model)-preloaded hydrogels (Gel-1-T, Gel-2-T, Gel-3-T and Gel-4-T) was investigated against *S. aureus* (as a bacterial model) for the potential application as wound dressings. The hydrogels were immersed in the bacterial suspension and the suspension was withdrawn for the agar assay study after 4, 8 and 24 h separately. As shown in Figure 10a, the photographs of agar plates displayed the colony units became fewer with the increasing of treating time for the same hydrogel category, illustrating the triclosan was gradually released from the hydrogels. After 8 h, the hydrogels all showed an excellent bacteria inhibition property, especially Gel-1-T, and there were almost no colony units for all the hydrogels after 24 h of treatment. Besides, the bacterial growth inhibition of Gel-1-T, Gel-2-T, Gel-3-T and Gel-4-T groups at 4 and 8 h was also studied by the colony-counting method and the untreated bacteria suspensions were used as control. As shown in Figure 10b, it could be found that Gel-1-T showed the best antibacterial property, achieving 2.40 and 4.78 colony-forming unit (CFU) log reduction of *S. aureus* after 4 h and 8 h, respectively, which corresponded to promising efficacies of 99.600% and 99.998%, respectively. Gel-4-T displayed the lowest killing bacteria capacity which corresponded to promising efficacies of 95.000% and 99.988% at 4 and 8 h, respectively. As a result, the bacteria-killing speed of hydrogels varied in order of Gel-1-T > Gel-2-T > Gel-3-T > Gel-4-T, which was corresponded with the drug release profile of hydrogels in Figure 7. However, there is a contradiction between the bacteria-killing speed (drug release rate) and cytotoxicity; for triclosan-preloaded hydrogels, the faster the drug is released, the higher the cytotoxicity is and the better antibacterial effect can be obtained. In general, the present study suggested that the peptide-based bis-acrylate/AAc hydrogels as drug carriers can be used as antibacterial wound dressings.

## 3. Materials and Methods 

### 3.1. Materials

*N*-Fluorenyl-9-methoxycarbonyl (FMOC)-protected L-amino acids (Fmoc-Gly-OH, Fmoc-Lys(Boc)-OH) and 2-chlorotrityl chloride resin (loading: 1.20 mmol/g) were purchased from GL Biochem Ltd. (Shanghai, People’s Republic of China). *N,N*-Diisopropyl ethylamine (DIEA), ninhydrin, *N,N,N,N*-tetramethyl-O-(1H-benzontriazol-1-yl) uranium hexafluorphosphate (HBTU), 1-Hydroxybenzontriazole (HOBt), trifluoroacetic acid (TFA), triisopropylsilane (Tis), dicyclohexylcarbodiimide (DCC), 4-dimethylaminopyridine (DMAP), and *N*-hydroxysuccinimide (NHS) were purchased from Alfa Aesar (Ward Hill, MA, USA). Succinic anhydride (SA), ethanolamine (EA), *N,N*-dimethylformamide (DMF), *N,N*-dimethyacetamide (DMAc), piperidine and triethylamine (TEA) were purchase from Sigma-Aldrich (Burlington, MA, USA).

### 3.2. Synthesis of Peptide-Based Bis-Methacrylate

The synthesis of the peptide-based crosslinker is illustrated in Figure 2. The peptide sequence Lys-Gly-Gly-Gly-Gly-Gly-Gly-Gly was synthesized using the solid-phase synthesis method. In brief, 2 g 2-chlorotrityl chloride resin was first immersed in DMF (20 mL) for 30 min. After draining off DMF solution, the FMOC protected amino acid (4 equiv relative to resin loading) and DIEA (6 equiv) in DMF (20 mL) were added and reacted for 1.5 h at room temperature. After that, the resin was washed with DMF for three times. Then, 20% piperidine/DMF (*v*/*v*) solution was added for 30 min to remove the FMOC-protected groups, and the resin was washed with DMF for three times. The presence of free amino groups was indicated by a blue color in the Kaiser test [41]. Thereafter, a DMF solution of the mixture of FMOC-protected amino acid (4 equiv), HBTU (4 equiv), HOBt (4 equiv), and DIEA (6 equiv) was added. After shaking for 1.5 h at room temperature, the reaction solution was drained off, and the resin was washed with DMF (three times). The absence of free amino groups was checked by 1% ninhydrin/methanol (*v*/*v*). After repetition of the condensation reaction and deprotection, the product on the resin could be obtained. Finally, the peptide-based crosslinker was synthesized directly on the peptidyl resin by coupling AAc to the amine groups. In detail, AAc (2 equiv), HOBt (4 equiv) and DCC (2.2 equiv) in DMF (3 mL) were added and reacted with peptidyl resin for 6 h at room temperature. The previous coupling reaction was repeated once more. Next, the resin was treated with a solution composed of 95% TFA, 2.5% TIPS and 2.5% water for 2 h to cleave the peptide crosslinker from the resin. The mixture was poured into cold ether and kept at −18 °C overnight, and then centrifuged. The supernatant was decanted and freeze-dried.

### 3.3. Preparation of Peptide-Based Bis-Acrylate/AAc Hydrogel

First, AAc and peptide-based bis-methacrylate with different feed ratios were dissolved in deionized water (DI water). The hydrogel was formed by radical polymerization using APS as an initiator and TEMED as an accelerator. The feed ratio of monomers was shown in in Table 1. After hydrogel formation, the hydrogels were immersed into 5% NaOH solution to neutralize the unreacted AAc. Subsequently, the hydrogels were immersed in excess DI water which was changed for several times for 3 days to leach out the impurities. In particular, to prepare the triclosan-preloaded hydrogels, 5 mg triclosan was dispersed in the hydrogel precursors and then formed by radical polymerization using the same method.

### 3.4. ^1^H NMR and FTIR Spectra

The ^1^H NMR spectra of peptide and peptide-based bis-acrylate were characterized using DMSO-d6 as solvents (Mercury VX-300 spectrometer, Varian, Palo Alto, CA, USA). FTIR spectra of freeze-dried hydrogels were characterized using the KBr method (AVATAR 360 spectrometer, Nicolet, Madison, WI, USA).

### 3.5. Morphology of Hydrogels

The morphologies of freeze-dried peptide-based bis-acrylate/AAc hydrogels were observed by SEM (S4500, Hitachi, Mountain View, CA, USA). Before SEM observation, the hydrogels at swelling equilibrium were quickly frozen using liquid nitrogen and then freeze-dried at −48 ^°^C for 48 h [3,10,32]. The pore sizes of freeze-dried hydrogels were measured from SEM images by Photoshop Software CS6 (Adobe, San Jose, CA, USA) to obtain the average pore size (n = 50).

### 3.6. Swelling Ratio of Hydrogels

The pH sensitivity of peptide-based bis-acrylate/AAc hybrid hydrogels was studied by immersing the freeze-dried hydrogels in buffer solutions with different pH values (pH 3, 7, 10 and 0.2 M) for 24 h. The buffer solutions were adjusted using HAc and NaAc for pH 3 buffer solution, and Na_2_CO_3_ and NaHCO_3_ for pH 10 buffer solution, and the sodium chloride was used to adjust the ionic strength. The swelling ratio was calculated according to Equation 1 (n = 3):
(1)
$$\mathrm{Swelling}\;\mathrm{ratio}=\frac{W_{t}-W_{d}}{W_{d}}$$

where W_t_ is the weight of the swollen hydrogel at time t (30, 60, 90, 120, 150, 180,210, and 240 min) and W_d_ is the dried weight of the freeze-dried hydrogel.

### 3.7. In Vitro Biodegradability of Hydrogels

The in vitro biodegradability of peptide-based bis-acrylate/AAc hydrogels was determined by weight loss (n = 3) and SEM observation. The swollen hydrogels were put into 50 mL PBS solution (pH: 7.4, ionic strength: 0.2 M) with trypsin (0.1 mg/mL), and the hydrogels were taken out and freeze-dried at predetermined time (1, 2, 3, 4, 5, 6, 7 and 8 days) for measuring the weight changes at 37 ^°^C and SEM observation.

### 3.8. Compressive Modulus of Hydrogels

The compressive modulus was measured (DMA Q800, TA Instruments, New Castle, DE, USA) with the maximum force of 0.01 N and the rate of 0.02 N × min^-1^ [3,32]. The hydrogels at swelling equilibrium were cut into the same size (height: 10 mm, diameter: 8 mm) for test. The modulus was calculated from the plot of the compressional stress versus strain.

### 3.9. In Vitro Drug Release from Hydrogels

The release profile of triclosan from triclosan-preloaded hydrogels was tested in the ultraviolet-visible spectrophotometer (Hitachi U-4100, Mountain View, CA, USA). The measurement process was described in a previous study [6]. In brief, the hydrogel was immersed into 200 mL solutions at 37 ^°^C. Two microliters of solutions were subsequently withdrawn and 2 mL fresh PBS solutions were added back to the vial. The amount of triclosan released was calculated by measuring the absorption at the wavelength of 282 nm.

### 3.10. Biocompatibility Evaluation

The attachment and proliferation of the L929 Fibroblast cells (ATCC, Washington, DC, USA) on the hydrogel surface was evaluated. In brief, the hydrogels (height: 10 mm, diameter: 8 mm) were placed into cell culture plates. Then, fibroblast cells were seeded (30,000 cells per well) and incubated for 24 and 48 h separately. The MTS solution was added and incubated for 3 h, and the cytotoxicity was evaluated by MTS assay according to the literatures [3,6]. Besides, in vivo toxicity was tested using a mouse model (6 week-old, male, C57BL/6). The mice were implanted with the swollen and sterile hydrogels with the same size (height: 5 mm, diameter: 5 mm) [3,42]. The mice were sacrificed, and major organs such as liver, spleen, kidney, heart and lung were fixed in 4% paraformaldehyde buffer solution for further hematoxylin and eosin (H&E) histological analysis. The experiments were approved and done in accordance with protocols approved by the experimental animal center of Tongji University, Shanghai, China. All care and handling of the animals were performed with the approval of Institutional Authority for Laboratory Animal Care.

### 3.11. Antibacterial Assessment

The *S. aureus* bacterial suspensions were prepared (1 × 10^5^ cfu/mL), then and the hydrogel was immersed and incubated at 37 ^°^C. At a predetermined time (0, 4, 8, and 24 h), 1 mL suspensions and the diluted suspensions were added into the agar plates and incubated at 37 °C for 24 h. The formed colony units were counted for the log reduction calculation by Equation 2 and the corresponding killing efficacy was calculated as Equation 3:
(2)
log reduction = log(colony units of control)−log(colony units of hydrogel)


(3)
$$\%\mathrm{kill}=\;\frac{\left(\mathrm{unit}\;\mathrm{count}\;\mathrm{of}\;\mathrm{control}-\mathrm{survivor}\;\mathrm{count}\;\mathrm{of}\;\mathrm{control}\right)}{\mathrm{unit}\;\mathrm{count}\;\mathrm{of}\;\mathrm{control}}\times 100$$


### 3.12. Statistical Analysis

All data were obtained with at least three repeated experiments and expressed as the average value ± standard deviation. In this study, * indicates significant difference (*p* < 0.05); ** indicates significant difference compared with all other conditions (*p* < 0.01). Statistical significance was calculated using the Student’s *t*-test.

## 4. Conclusions

In this study, a series of biodegradable peptide-based bis-acrylate/AAc hydrogels was designed and synthesized. The hybrid hydrogels showed a pH-dependent swelling ratio and the swelling ratio would increase when the solution became alkaline. The biodegradation of hydrogels was tested and the results proved the hydrogels could almost biodegrade after 8 days in the presence of trypsin. The mechanical and cytotoxicity properties of the hydrogels were also tested and the hydrogels displayed integrated advantages of tough and non-toxic property. Finally, triclosan was preloaded during the hydrogel formation for drug release and antibacterial studies. In summary, compared with the conventional biodegradable hydrogels, this pH-sensitive and biodegradable peptide-based bis-acrylate/AAc hydrogels would have great potential for drug release and wound dressing applications.

## Figures and Tables

**Figure 1 molecules-23-03383-f001:**
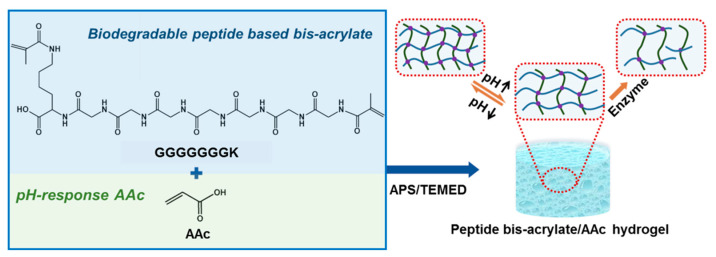
Schematic diagram of the peptide-based bis-acrylate/ acrylic acid (AAc) hydrogel with pH sensitivity and biodegradability. The hydrogel was formed by radical polymerization using ammonium persulfate (APS) as an initiator and tetramethylethylenediamine (TEMED) as an accelerator.

**Figure 2 molecules-23-03383-f002:**
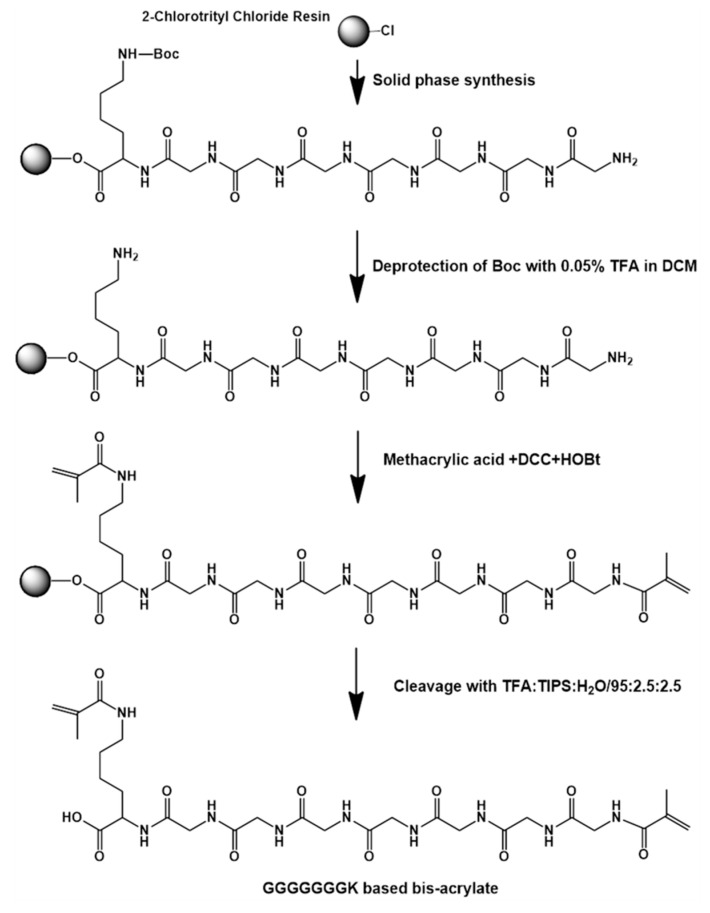
Schematic diagram of the peptide-based bis-acrylate as a crosslinker. The peptide sequence Lys-Gly-Gly-Gly-Gly-Gly-Gly-Gly was synthesized using the solid-phase synthesis method. The peptide-based crosslinker was synthesized directly on the peptidyl resin by coupling AAc to the amine groups.

**Figure 3 molecules-23-03383-f003:**
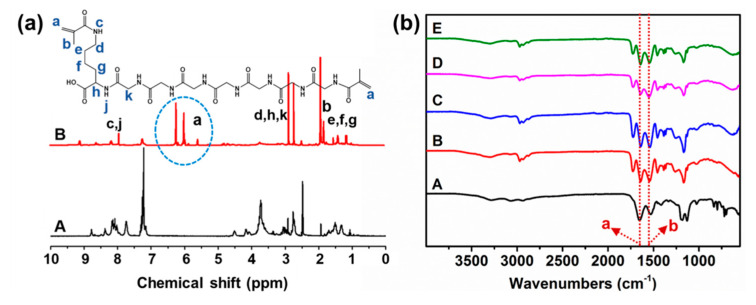
(**a**) ^1^H NMR of (A) the peptide GGGGGGGK (black line) and (B) the peptide-based bis-acrylate (red line). (**b**) FTIR of the hydrogels (A: peptide-based bis-methacrylate; B: Gel-1; C: Gel-2; D: Gel-3; E: Gel-4). Peak a: 1650 and 1540 cm^–1^, Peak b: 1540 cm^–1^.

**Figure 4 molecules-23-03383-f004:**
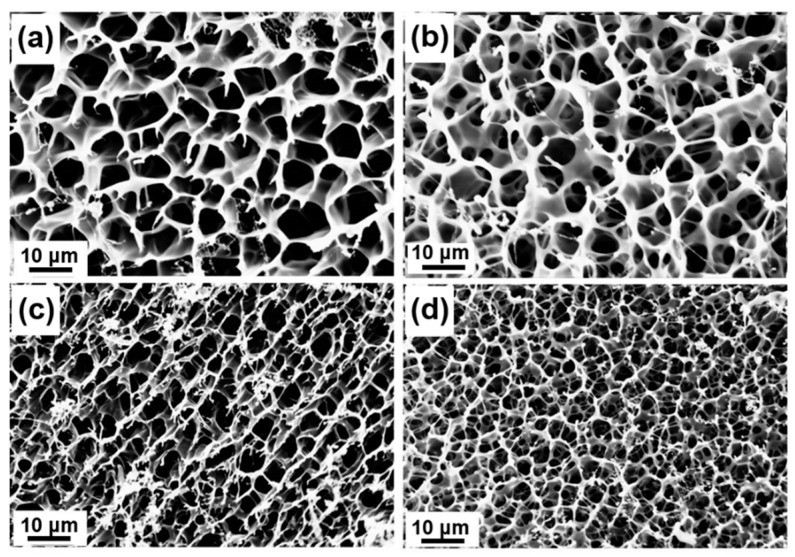
SEM images of homogeneous peptide-based bis-acrylate/AAc hydrogels before biodegradation: (**a**) Gel-1; (**b**) Gel-2; (**c**) Gel-3; (**d**) Gel-4. With the increasing of peptide-based bis-acrylate contents, the pore size of the hydrogels would decrease.

**Figure 5 molecules-23-03383-f005:**
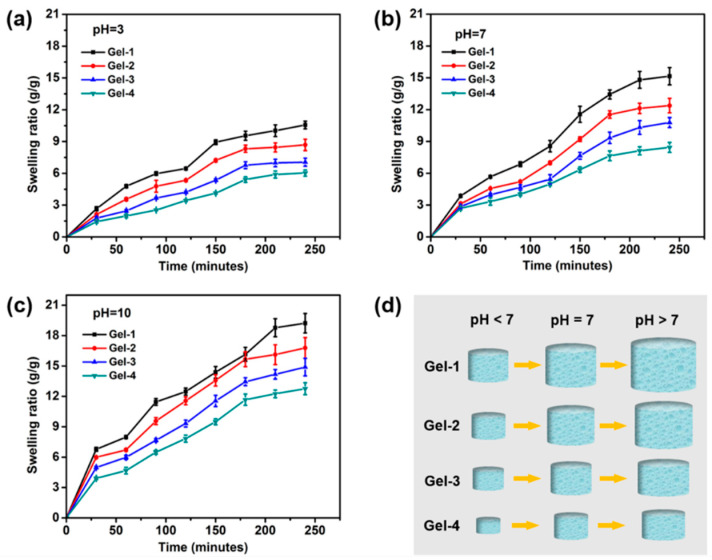
Swelling ratios of Gel-1 (black line), Gel-2 (red line), Gel-3 (blue line) and Gel-4 (green line) hydrogels at pH = 3 (**a**), pH = 7 (**b**) and pH = 10 (**c**) as a function of time. (**d**) The trend of swelling vs pH of each hydrogel. The swelling ratio would increase when the solution became alkaline.

**Figure 6 molecules-23-03383-f006:**
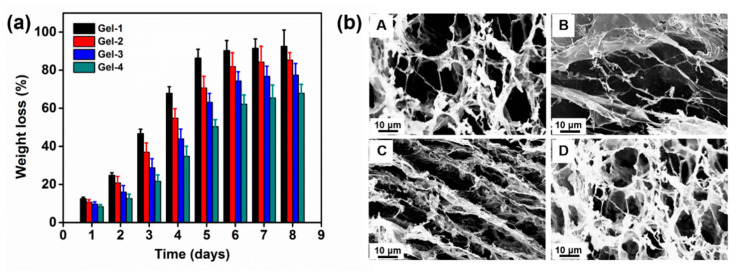
(**a**) Weight loss of the hydrogels in the presence of trypsin at a concentration of 0.1 mg/mL in the PBS solution as a function of time. Black bar: Gel-1; red bar: Gel-2; blue bar: Gel-3; green bar: Gel-4. (**b**) SEM of the hydrogel after 4 days biodegradation in the PBS at the trypsin concentration of 0.1 mg/mL (A: Gel-1; B: Gel-2; C: Gel-3; D: Gel-4).

**Figure 7 molecules-23-03383-f007:**
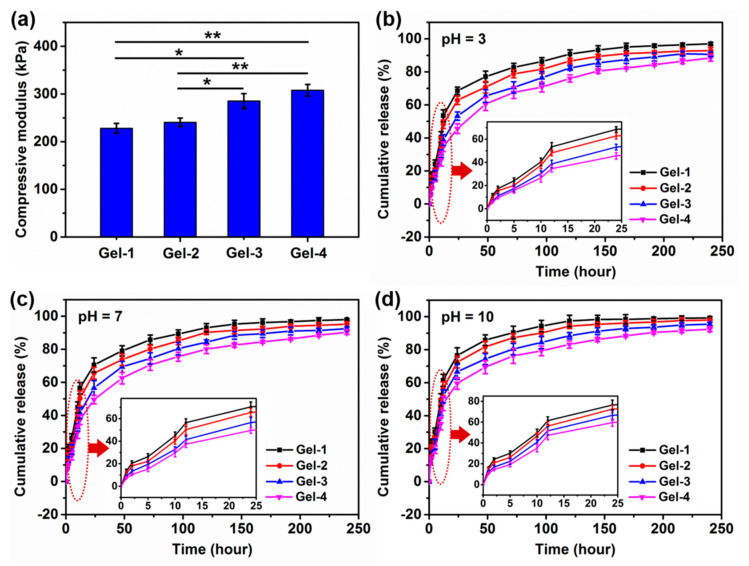
(**a**) Compression modulus of Gel-1, Gel-2, Gel-3 and Gel-4 in a neutral environment. Controlled release of triclosan from the hydrogels in various environments as a function of time: (**b**) pH = 3; (**c**) pH = 7; (**d**) pH = 10. Insets showed magnifying drug release profile in 25 h. * indicates significant difference (*p* < 0.05); ** indicates significant difference compared with all other conditions (*p* < 0.01). Statistical significance was calculated using the Student’s *t*-test.

**Figure 8 molecules-23-03383-f008:**
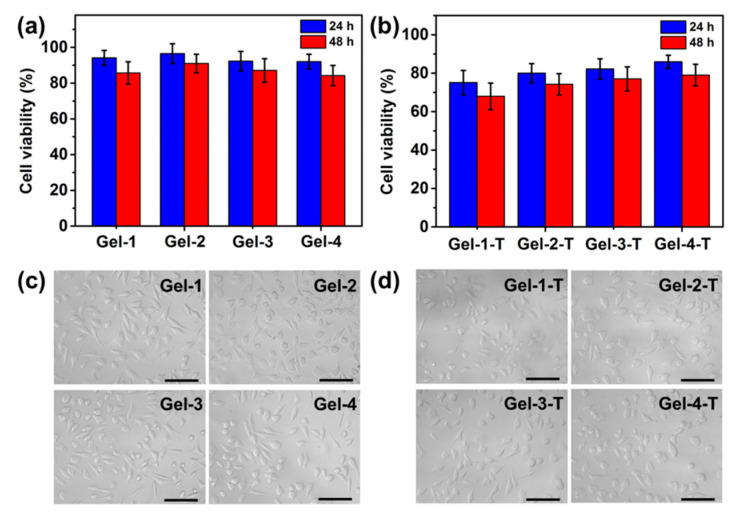
Cytotoxicity assay by the MTS method of (**a**) Gel-1, Gel-2, Gel-3 and Gel-4, and (**b**) triclosan-loaded Gel-1, Gel-2, Gel-3 and Gel-4 (labeled as Gel-1-T, Gel-2-T, Gel-3-T and Gel-4-T, respectively) after 24 h (blue bar) and 48 h (red bar). Cell images of the L929 cells in Dulbecco’s modified eagle medium (DMEM) after 24 h incubating with (**c**) Gel-1, Gel-2, Gel-3 and Gel-4, and (**d**) Gel-1-T, Gel-2-T, Gel-3-T and Gel-4-T. Scale bars are 100 μm.

**Figure 9 molecules-23-03383-f009:**
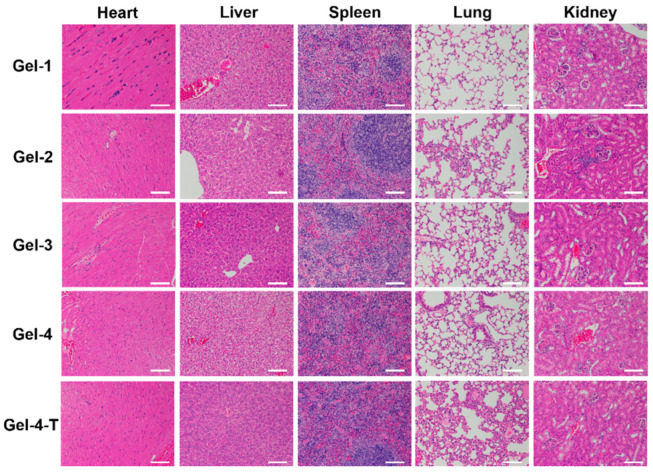
In vivo toxicity assessment of hydrogels. Hematoxylin-eosin (H&E) stained tissue slices (liver, spleen, kidney, heart and lung) of mice injected with hydrogels after 24 h (the white scale bar is 200 μm).

**Figure 10 molecules-23-03383-f010:**
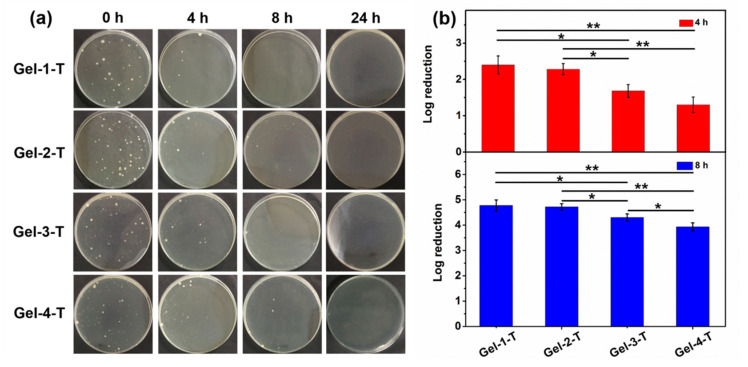
(**a**) Bacterial colonies formed using an agar diffusion assay treated with Gel-1-T, Gel-2-T, Gel-3-T and Gel-4-T against *S. aureus* at 0, 4, 8 and 24 h. (**b**) The bacterial growth inhibition of Gel-1-T, Gel-2-T, Gel-3-T and Gel-4-T at 4 (red bar) and 8 h (blue bar) investigated by the colony-counting method. * indicates significant difference (*p* < 0.05); ** indicates significant difference compared with all other conditions (*p* < 0.01). Statistical significance was calculated using the Student’s *t*-test.

**Table 1 molecules-23-03383-t001:** The feed ratios of the peptide-based bis-acrylate/AAc hybrid hydrogels.

Component	Gel-1	Gel-2	Gel-3	Gel-4
Peptide-based bis-acrylate (mg)	40	60	80	100
AAc (mg)	360	340	320	300
APS (mg)	20	20	20	20
TEMED (μL)	10	10	10	10
DI water (mL)	3	3	3	3

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
