# Peer review of "Biodegradable and pH Sensitive Peptide Based Hydrogel as Controlled Release System for Antibacterial Wound Dressing Application"

_molecules, 2018, doi:10.3390/molecules23123383_

Round 1

Reviewer 1 Report

The manuscript is focused on the investigation of novel pH-responsive hydrogels able to release drug at specific pH in wound healing application.

The manuscript has to be improved in the different sections as described in the detailed comments reported here below.

Detailed comments

line 27: “porous structure”  better explain if you mean pores due to the 3D macromolecular network or physical pores

line 28: “moisture”  better explain what you mean by that

line 29: “diverse biomedical applications”  give examples for that

lines 30-31: “ECM-like  better explain

line 31: “rehydrating dead tissues”  if the tissue is “dead”, explain the reason for a rehydration of it

line 40: “commonly”  give at least two references about it

line 41: “... infected and fluid ...”  better explain what you mean by “fluid site”

line 42: “slight alkaline”  better explain the reason for an alkaline environment

line 43: “exudation of infected wound”  you have to better introduce this problem

line 44: “biodegradable hydrogels”  you did not mention any classification for hydrogels

lines 46-47: better explain this point

lines 48-49: you have to better explain what you mean, giving examples of hydrogels with these characteristics of degradability and pH-responsiveness

line 50: “peptide based materials”  you have to explain if they are hydrogels or not

lines 52-53: “are also non-toxic”  it depends on the hydrogel formulation. Give a reference

line 54: “enzymes”  you have to explain the role of the enzymes

line 55: “can be designed”  you have to summarize the requirement of the hydrogel for that specific application

lines 62-63: “the interior morphology”  what do you mean by that?

lines 66-67: conclusions are not required in the Introduction section

line 82: give a reference for the attribution of these bands (FTIR)

line 84: “this crosslinker”  you have to give a reference or results for that

line 88: “... would ne more ...”  as you tested the biodegradation of your hydrogels, you have to compare these data to literature

line 101: you have to better explain the relationship between the pore size and the crosslinking density. Moreover, you have to report the crosslinking density. Porous structure is related to the freeze-drying treatment; you have to better explain if you want to use freeze-dried hydrogels as drug release pH-responsive systems

line 106: “pore size”  describe how you obtain the data of pore dimensions

line 118: you have to better explain how you can balance the absorption of the exudate and the release of drug

line 118: “water uptake capacity”  did you test freeze-dried samples or compact ones?

line 118: “essential properties”  give a reason for that

lines 121-124: it should be better to add graphs that show the trend of swelling vs pH for each prepared hydrogels

Figure 4: you have to report the swelling ratio using the same maximum y-axis value

line 144: “significant ...”  do you mean p < 0.05?

line 163: “average pore size”  you have to explain how you detect pores in these structures (Figure 5 (b))

line 172: “wound dressing”  for this specific application, it is important the compressive behaviour as well as the shear behaviour. You have to take it into account

line 173: “neutral environment”  you have to explain why you tested the hydrogels in neutral condition if you have to use them in alkaline environment

line 180: “will be weakened”  you have to test the hydrogels in this condition, as well as in acidic condition. Otherwise, what you state is only a hypothesis

line 212: “pore size”  it is unclear what you by pores; are they the ones you obtained by freeze-drying? Please better explain this point

lines 217-218: as already reported, this is a hypothesis that needs to be confirmed by experimental data. Otherwise, you have to better explain your hypothesis

line 228: “... significant ...”  do you mean p < 0.05?

cytotoxicity  you have to better explain these results. If you detected a cytotoxic effect due to triclosan, how can you apply the hydrogel on a damaged tissue (i.e., as wound dressing)?

line 245: you have to highlight the possible correlation among these parameters

what about the in vivo toxicity of the loaded hydrogels?

line 262: you have to discuss how you can match the results related to the loaded hydrogels as antibacterial material with the cytotoxicity of the released drug

lines 312-313: why did you use another technique to verify the presence/absence of free amino group? give a reference for the method you used

line 328: you have to report number of replicates and specimen dimensions

line 332: “interior morphologies”  as already reported, you have to better explain what you mean by that. In fact, the porous morphology you detected by SEM is related to the freeze-drying technique not to the hydrogel itself

line 333: “before test”  do you mean before SEM analysis?

line 336: better explain the composition of the solution you used

line 338: detail the considered time points

line 343: detail the considered time points

line 344: “maximum force”  better explain what you mean as maximum force

line 347: you have to better explain the method used to load the hydrogels with triclosan; did you investigate the quantity of drug loaded in the hydrogels?

in vivo tests: you have to explain which is the rationale for that experiment

line 358: report the number of the authorization for the in vivo experiment

line 366: you need to perform statistical analysis to check possible statistical differences among the different hydrogels

lines 380-382: is it a refuse? please revise this part

Author Response

Response to Reviewer

Reviewer 1#

The manuscript is focused on the investigation of novel pH-responsive hydrogels able to release drug at specific pH in wound healing application.

The manuscript has to be improved in the different sections as described in the detailed comments reported here below.

Response summary: Thank the reviewer for the valuable comments. As the reviewer suggested, we have reorganized the Introduction section to provide an appropriate context and background of previous works of pH-responsive and biodegraded hydrogels. We have improved Figure 1&8, checked the statistical differences of hydrogels, and had more detailed discussion in the revised manuscript. Besides, we have added more information to the clearly explain Materials and Methods. Finally, the revised manuscript has been carefully checked to avoid any inappropriate terms, typos and other mistakes to improve the clarity and readability. We hope that the reviewer finds our responses satisfactory and convincing. The responses to comments are listed below:

Detailed comments

1. Line 27: “porous structure” better explain if you mean pores due to the 3D macromolecular network or physical pores.

Response: Thank the reviewer for the comment. The “porous structure” is referred to the pores due to the macromolecular network as the reviewer mentioned. We have changed this sentence to “Hydrogels with three-dimensional, interconnected and polymer networks have the advantages of high water absorption capacity and biocompatibility which can be investigated for diverse biomedical applications including tissue engineering, axonal regeneration, wound dressings and controlled drug release” (line 27).

2. Line 28: “moisture” better explain what you mean by that.

Response: Thank the reviewer for the comment. As the hydrogels are capable of holding large amounts of water in their three-dimensional networks, the hydrogels keep moist compared with other materials (J. Adv. Res., 2015, 6, 105). In order to explain it more clearly, we have changed this sentence to “Hydrogels with three-dimensional, interconnected and polymer networks have the advantages of high water absorption capacity and biocompatibility which can be investigated for diverse biomedical applications including tissue engineering, axonal regeneration, wound dressings and controlled drug release” (line 28).

3. Line 29: “diverse biomedical applications” give examples for that.

Response: Thank the reviewer for the comment and we have added the examples as the reviewer suggested. We have changed this sentence to “Hydrogels with three-dimensional, interconnected and polymer networks have the advantages of high water absorption capacity and biocompatibility which can be investigated for diverse biomedical applications including tissue engineering, axonal regeneration, wound dressings and controlled drug release” (lines 29-30).

4. Lines 30-31: “ECM-like better explain.

Response: Thank the reviewer for the comment. We have improved this sentence to “hydrogels with the crosslinked structure similar to native extracellular matrix (ECM) are promising candidates for wound dressings because hydrogels can create a moist environment, absorb wound fluids and facilitate autodebridement of wounds by rehydrating slough and enhancing the rate of autolysis” (lines 30-33).

5. Line 31: “rehydrating dead tissues” if the tissue is “dead”, explain the reason for a rehydration of it.

Response: Thanks for the reviewer’s comment. Based on the literature (Amer. J. Perinatol., 2004, 21, 409), we have revised this sentence to “hydrogels with the crosslinked structure similar to native extracellular matrix (ECM) are promising candidates for wound dressings because hydrogels can create a moist environment, absorb wound fluids and facilitate autodebridement of wounds by rehydrating slough and enhancing the rate of autolysis” (lines 30-33).

6. Line 40: “commonly” give at least two references about it

Response: Thanks for the reviewer’s comment. According to the reviewer’s comment, we have added several references here (Colloid. Surface. B. 2011, 88, 593; J. Mater. Chem. B, 2013, 1, 5578; Eur. Polym. J., 2018, 101, 282) (line 42).

7. line 41: “... infected and fluid ...”  better explain what you mean by “fluid site”

Response: Thanks for the reviewer’s comment. We have improved this sentence to give an appropriate example here based on the reference (Arch. Dermatol. Res., 2007, 298, 413). The sentence has been changed to “the chronic wounds and infected wounds with a high bacterial load show a slight alkaline pH (above 7.3), therefore, the hydrogels will release drugs at a faster rate to kill the bacteria in a short period; in contrast, wounds with pus or necrotic tissue (such as ulcers) show an acidic pH so that the hydrogels will release the drugs in a slower and more steady profile” (lines 42-46).

8. Line 42: “slight alkaline” better explain the reason for an alkaline environment.

Response: Thanks for the reviewer’s comment. As the pH value in wounds is influenced by many different endogenous and exogenous factors, and different types of wounds show different pH (Arch. Dermatol. Res., 2007, 298, 413). We have improved this sentence to given an appropriate example here based on the reference (Arch. Dermatol. Res., 2007, 298, 413). The sentence has been changed to “the chronic wounds and infected wounds with a high bacterial load show a slight alkaline pH (above 7.3), therefore, the hydrogels will release drugs at a faster rate to kill the bacteria in a short period; in contrast, wounds with pus or necrotic tissue (such as ulcers) show an acidic pH so that the hydrogels will release the drugs in a slower and more steady profile” (lines 42-46).

9. Line 43: “exudation of infected wound” you have to better introduce this problem.

Response: Thanks for the reviewer’s comment. According to the reviewer’s comment, we have added a specific wound type here for introduce this problem clearly. The sentence has been changed to “the chronic wounds and infected wounds with a high bacterial load show a slight alkaline pH (above 7.3), therefore, the hydrogels will release drugs at a faster rate to kill the bacteria in a short period; in contrast, wounds with pus or necrotic tissue (such as ulcers) show an acidic pH so that the hydrogels will release the drugs in a slower and more steady profile” (lines 42-46).

10. Line 44: “biodegradable hydrogels” you did not mention any classification for hydrogels.

Response: Thanks for the reviewer’s comment. We realized this transition of introducing “biodegradable hydrogels” was not very smooth, thus we have rewritten this section in a separate paragraph (lines 47-54). The revised paragraph is shown below.

Compared with non-degradable hydrogels, the biodegradable hydrogels which can degrade into safe molecules are more coincident with the requirements of biomedical applications as they can be introduced into the bodies with minimally invasion (ACS Macro Lett., 2012, 1, 409). Besides, as a drug release system, the release rate of drugs from biodegradable hydrogels can be controlled by several factors such as the enzyme concentration and crosslink density (J. Control. Release, 1997, 44, 237; Adv. Drug Deliver. Rev., 2004, 56, 1621; J. Mater. Chem. B., 2014, 2, 6660). Moreover, the hydrogel degradation can allow for hydrogel clearance after drug exhaustion (J. Control. Release, 1997, 44, 237; ACS Appl. Mater. Inter. 2015, 7, 14338). Herein, we anticipate to construct a hydrogel combined pH-sensitivity and biodegradation property for controlled drug release and wound dressing application (Polymer, 2009, 50, 4308; J. Control. Release, 2014, 193, 214; Sci. Rep., 2016, 6, 29978).

11. Lines 46-47: better explain this point.

Response: Thanks for the reviewer’s comment. We have added some detailed factors to give a clearer explanation. The sentence has been revised to “Besides, as a drug release system, the release rate of drugs from biodegradable hydrogels can be controlled by several factors such as the enzyme concentration and crosslink density (J. Control. Release, 1997, 44, 237; Adv. Drug Deliver. Rev., 2004, 56, 1621; J. Mater. Chem. B., 2014, 2, 6660)” (lines 49-51).

12. Lines 48-49: you have to better explain what you mean, giving examples of hydrogels with these characteristics of degradability and pH-responsiveness.

Response: Thanks for the reviewer’s comment. We have revised the sentence and added several references to improve the readability (lines 52-54). The revised paragraph is shown below.

Herein, we anticipate constructing a hydrogel combined pH-sensitivity and biodegradation property for controlled drug release and wound dressing application (Polymer, 2009, 50, 4308; J. Control. Release, 2014, 193, 214; Sci. Rep., 2016, 6, 29978).

13. Line 50: “peptide based materials” you have to explain if they are hydrogels or not.

Response: Thank the reviewer for the comment. We have changed the “peptide based materials” to “peptide based hydrogels” (lines 55, 57 and 58).

14. Lines 52-53: “are also non-toxic” it depends on the hydrogel formulation. Give a reference.

Response: Thank the reviewer for the comment and we have improved this sentence and added the relevant reference in the revised manuscript (line 58).

15. Line 54: “enzymes” you have to explain the role of the enzymes.

Response: Thank the reviewer for the comment and we have added two specific enzymes for example and their roles in the revised manuscript (lines 59-60). The sentence has been revised to “Most importantly, the degradation rate of the peptide based hydrogels can be careful tuned by the monomer design and the concentration of enzymes such as trypsin and α-chymotrypsin for catalyzing the hydrolysis of amide bonds” (Biomaterials, 2000, 21, 1499; J. Appl. Polym. Sci. 2010, 117, 3386).

16. Line 55: “can be designed” you have to summarize the requirement of the hydrogel for that specific application.

Response: Thank the reviewer for the comment. We have rewritten this sentence to give a better conclusion of this paragraph. The sentence has been revised to “Considering these effects, endowing the biodegradable peptide based hydrogels with pH-response property can be designed for drug release and wound dressing application” (lines 60-62).

17. Lines 62-63: “the interior morphology” what do you mean by that?

Response: Thank the reviewer for the comment. We used the “interior morphology” to represent the interior porous structure for hydrogels which was also appeared in the previous studies (Carbohyd. Polym., 2007, 67, 491; Chem. Eng. J., 2010, 159, 247). In the revised manuscript, we have revised the sentence to “The morphologies of the hybrid hydrogels were observed by scanning electron microscopy (SEM)” (line 69).

18. Lines 66-67: conclusions are not required in the Introduction section.

Response: Thank the reviewer for the comment and we have deleted the conclusion in the revised manuscript.

19. Line 82: give a reference for the attribution of these bands (FTIR).

Response: Thank the reviewer for the comment and we have added references to support the conclusion (line 87).

20. Line 84: “this crosslinker” you have to give a reference or results for that.

Response: Thank the reviewer for the comment. The description of “this crosslinker” is referred to the peptide based bis-acrylate prepared in this study. In order to avoid confusion, we have improved the sentence to “Benefited from this design, the peptide based bis-acrylate, as a crosslinker, has the advantages of biocompatibility and enzymatic biodegradation” (lines 89-90).

21. Line 88: “... would ne more ...”  as you tested the biodegradation of your hydrogels, you have to compare these data to literature.

Response: Thank the reviewer for the comment. According to the reviewer’s comment, we have compared the results in the revised manuscript. In order to make the context more coherent, we have moved this section to the “Enzymatic Biodegradation of Hydrogels” section (lines 167-173). Xu et al., prepared an injectable hydrogel based on two recombinant proteins (ULD-TIP1 and ULD-GGGWRESAI) and found this hydrogel could exhibit a relatively quick degradation rate even in the pure PBS buffer (Int. J. Biol. Macromol., 2017, 95, 294). Their studies showed the hydrogel was almost dissolved and/or degraded after 144 h in pure PBS solution which was too fast to meet requirements in the practical application. Compared with Xu’s work, the hydrogels in our work showed a more controllable and longer degradation process more than 8 days which was depended on the crosslinking density and enzyme concentration (Acta biomater., 2018, 65, 305).

22. Line 101: you have to better explain the relationship between the pore size and the crosslinking density. Moreover, you have to report the crosslinking density. Porous structure is related to the freeze-drying treatment; you have to better explain if you want to use freeze-dried hydrogels as drug release pH-responsive systems

Response: Thank the reviewer for the comment. We have added more information on the crosslinking density in the Supporting Information. The crosslinking density (1/MC) could be estimated by Flory-Rehner’s equation (Equation 1 and 2) as shown below (J. Appl. Polym. Sci., 1992, 46, 783; Int. J. Polym. Sci., 2011, 343062.).

                                                                     (1)

                       (2)

In this equation, MC is average molar mass between the network crosslinks, QV is the volume swelling ratio of hydrogels, v is the polymer volume, V is the molar volume of H2O, x is the Flory interaction parameter between a solvent and a polymer, ρ is the polymer density and ρs is the density of H2O. The estimation crosslinking density result of hydrogels (pH = 7) is shown below.

Table S1. Calculation results of crosslinking density of hydrogels (pH = 7).

Hydrogel

Crosslinking density (10-5)

Gel-1

5.2

Gel-2

5.7

Gel-3

6.0

Gel-4

6.8

Besides, we used the swollen hydrogels for the drug release investigation in this study, and found the release profile was dependent on the pore size. As the pore size increased, the drug release rate would decrease which was consistent with the previous studies (Acta biomater., 2018, 65, 305; ACS Appl. Mater. Inter. 2018, 10, 13304).

23. Line 106: “pore size” describe how you obtain the data of pore dimensions

Response: Thank the reviewer for the comment. The pore sizes of freeze-dried hydrogels in this study were measured from SEM images by Photoshop Software to obtain the average pore size (n = 50). We have added the method in the “Materials and Methods” section (lines 355-357) and the statistics results have been added in the revised Supporting Information (Figure S1).

Figure S1. The pore size and size distribution measured from the SEM images of freeze-dried hydrogels.

24. Line 118: you have to better explain how you can balance the absorption of the exudate and the release of drug.

Response: Thank the reviewer for the comment. We have added more information concerning the water uptake and drug release function of hydrogels as wound dressing (lines 121-122, 220-224). The swelling ratio was relevant to the pore structure of hydrogels in this study; the hydrogel with a denser structure would have a lower swelling ratio. The swelling ratio could be also affected by the pH value of solutions; the hydrogels would have a better water absorption ability in the alkaline environment. As a drug release system, the hydrogels with a denser structure or in an acid condition would perform a slower release profile, which was in positive correlation with the swelling ratio.

25. Line 118: “water uptake capacity” did you test freeze-dried samples or compact ones?

Response: Thank the reviewer for the comment. The swelling ratio, as an index to assess the water uptake capacity, is characterized by immersing the freeze-dried hydrogels into PBS and the weight of hydrogels at a predetermined time is measured. We have improved the method of swelling ratio characterization in the “Materials and Methods” section (line 358-365).

26. Line 118: “essential properties” give a reason for that.

Response: Thank the reviewer for the comment. We have added several references to give an appropriate reason for that (Adv. Funct. Mater., 2014, 24, 3933; Acta Biomater., 2016, 38, 59; J. Mater. Chem. B., 2017, 5, 8975) (line 121).

27. Lines 121-124: it should be better to add graphs that show the trend of swelling vs pH for each prepared hydrogels.

Response: Thank the reviewer for the comment. According to the reviewer’s comment, we have added a figure (revised Figure 4d) to show the trend of swelling vs pH. The revised Figure 4 is shown below.

Figure 4. Swelling ratio of Gel-1, Gel-2, Gel-3 and Gel-4 hydrogels in (a) pH = 3, (b) pH = 7 and (c) pH = 10. (d) The trend of swelling vs pH of each hydrogel.

28. Figure 4: you have to report the swelling ratio using the same maximum y-axis value.

Response: Thank the reviewer for the comment. We have reset the y-axis using the same maximum values. The revised Figure 4 is shown below.

Figure 4. Swelling ratio of Gel-1, Gel-2, Gel-3 and Gel-4 hydrogels in (a) pH = 3, (b) pH = 7 and (c) pH = 10. (d) The trend of swelling vs pH of each hydrogel.

29. Line 144: “significant ...” do you mean p < 0.05?

Response: Thank the reviewer for the comment. In the original manuscript, we meant all the hydrogels showed obvious weight loss which was more than 8% after 24 h. In order to avoid ambiguity, we have changed the “significant” to “obvious” (line 147).

30. Line 163: “average pore size” you have to explain how you detect pores in these structures (Figure 5 (b))

Response: Thank the reviewer for the comment. The pore sizes of freeze-dried hydrogels after biodegradation were also measured from SEM images by Photoshop Software. As the pore sizes increased largely and some of the walls in hydrogels would be lost with increasing of biodegradation time, the pore sizes of hydrogels showed great differences. We have deleted the specific sizes in the revised manuscript.

31. Line 172: “wound dressing” for this specific application, it is important the compressive behaviour as well as the shear behaviour. You have to take it into account.

Response: Thank the reviewer for the comment. We agree with the reviewer that the mechanical properties are important for the practical application. We will take a comprehensive investigation of the mechanical properties in the following researches.

32. Line 173: “neutral environment” you have to explain why you tested the hydrogels in neutral condition if you have to use them in alkaline environment.

Response: Thank the reviewer for the comment. In this study, the prepared hydrogels were immersed into 5% NaOH solution to neutralize the unreacted AAc. Subsequently, the hydrogels were immersed in excess DI water to reach swelling equilibrium. The hydrogels were then cut into the shape with the same size for mechanical test, and in this case, the hydrogels were neutral. We have improved the Introduction section on the pH values of wound sites (lines 42-46) and showed the pH value in wounds is a dynamic factor that can change rapidly with therapeutic interventions (Arch. Dermatol. Res., 2007, 298, 413). For this reason, we conducted the compression test using the swollen hydrogels which were neutral. We have added the reason of this test and revised the sentence to “As the pH value in wounds can change with therapeutic interventions, the compressive property of the swollen and neutral peptide based bis-acrylate/AAc hydrogels was tested” (line 180-182).

33. Line 180: “will be weakened” you have to test the hydrogels in this condition, as well as in acidic condition. Otherwise, what you state is only a hypothesis.

Response: Thank the reviewer for the comment. In the original manuscript, the explanations were summarized from Li’s study. As we did not test the compression in acidic and alkaline conditions, we have deleted the discussion on the pH influence.

34. Line 212: “pore size” it is unclear what you by pores; are they the ones you obtained by freeze-drying? Please better explain this point.

Response: Thank the reviewer for the comment. The “pore size” in Shang’s study was observed by SEM using freeze-dried hydrogels (line 216). The hydrogels were immersed in DI water and quickly frozen using liquid nitrogen for 5 min. In this way, the natural hydrogel porous structure could be fixed, and their structure could be monitored by SEM. We have improved the sentence: “Shang et al., have confirmed that the pore size of freeze-dried AAc based hydrogel in alkaline buffer solution would turn to be larger than that in acidic condition” (lines 215-217).

35. Lines 217-218: as already reported, this is a hypothesis that needs to be confirmed by experimental data. Otherwise, you have to better explain your hypothesis.

Response: Thank the reviewer for the comment. According to the reviewer’s comment, we have improved the conclusion. When applied at the bacterial infected wounds (in slight alkaline condition), the peptide based bis-acrylate/AAc hydrogels will release drugs at a faster rate for killing the bacteria in a short period. When used at wounds with pus or necrotic tissue (in an acidic condition), the hydrogels will release the drugs in a slower profile, achieving a long-term drug release effect (lines 220-224).

36. Line 228: “... significant ...” do you mean p < 0.05?

Response: Thank the reviewer for the comment. We have performed detailed statistical analysis on cytotoxicity assay results using Student’s t-test between each group. The result showed the statistical significance in MTS results (Figure 7a&b) was higher than 0.05 between each group. We have added the results in the revised manuscript (line 233).

37. Cytotoxicity you have to better explain these results. If you detected a cytotoxic effect due to triclosan, how can you apply the hydrogel on a damaged tissue (i.e., as wound dressing)?

Response: Thank the reviewer for the comment. Compared with the pure peptide based bis-acrylate/AAc hydrogels, the triclosan preloaded hydrogels showed lower cell viability by MTS assay. We have improved the discussion on the cytotoxicity of triclosan preloaded hydrogels (line 237-249).

Besides, the cell viability of triclosan preloaded hydrogels (Gel-1-T, Gel-2-T, Gel-3-T and Gel-4-T) was also investigated as control. Seen from Figure 7b, the cell viability of triclosan preloaded hydrogels all displayed a bit of decrease after 24 and 48 h. For the Gel-4-T hydrogel, the cell viability was about 85.92% and 79.03% after 24 and 48 h, respectively, which did not generate severe cytotoxicity. In contrast, the Gel-1-T hydrogel showed 75.13% and 67.67% in cell viability after 24 and 48 h, respectively, which displayed the lowest biocompatibility among the four hydrogels. It was notably that the cell viability of the Gel-1-T, Gel-2-T, Gel-3-T and Gel-4-T hydrogels was related to the drug release profile; that is, the hydrogel with a higher release rate had a higher cytotoxicity caused by the released triclosan which might cause damage to cells. However, the results also indicated the cytotoxicity of drug preloaded hydrogels could be modulated by the feed ratio of peptide based bis-acrylate and AAc. As a result, the good biocompatibility indicated of the peptide based bis-acrylate/AAc hydrogels and the Gel-4-T hydrogel in this study were good candidates as wound dressings.

38. Line 245: you have to highlight the possible correlation among these parameters.

Response: Thank the reviewer for the comment. We have added the correlation among these parameters in the revised manuscript (lines 258-260). The revised sentences are shown below.

Besides, the cell viability of drug preloaded hydrogels could be affected by the crosslinking density of hydrogels; as the crosslinking density increased, the pore size of hydrogels would be decreased, leading the drug release rate to be slower.

39. What about the in vivo toxicity of the loaded hydrogels?

Response: Thank the reviewer for the comment. As the Gel-4-T hydrogel in this study was the best candidate for wound dressing application among the four triclosan preloaded hydrogels, we have characterized the in vivo toxicity of Gel-4-T hydrogel, and the results have been added in the revised manuscript (revised Figure 8).

Figure 8. In vivo toxicity assessment of hydrogels. H&E stained tissue slices (liver, spleen, kidney, heart and lung) of mice injected with hydrogels after 24 h (the white scale bar is 200 μm).

40. Line 262: you have to discuss how you can match the results related to the loaded hydrogels as antibacterial material with the cytotoxicity of the released drug.

Response: Thank the reviewer for the comment. According to the reviewer’s comment, we have added more discussion concerning the results of antibacterial assessment and the cytotoxicity in the revised manuscript (lines 298-301).

41. Lines 312-313: why did you use another technique to verify the presence/absence of free amino group? give a reference for the method you used.

Response: Thank the reviewer for the comment. In the synthesis of peptide, we used the Kaiser test (line 327) to verify the presence of free amino groups (Anal. Biochem., 1970, 34, 595). The principle of Kaiser test is that free terminal amino groups can be detected by ninhydrin/methanol (line 331) to show a color. In this test, the free amino groups could be indicated by a blue color. We have added the reference in the proper place (line 327).

42. Line 328: you have to report number of replicates and specimen dimensions.

Response: Thank the reviewer for the comment. We have added the number of replicates and specimen dimensions in the revised manuscript.

43. Line 332: “interior morphologies” as already reported, you have to better explain what you mean by that. In fact, the porous morphology you detected by SEM is related to the freeze-drying technique not to the hydrogel itself.

Response: Thank the reviewer for the comment. Freeze-drying is a common method for the morphology investigation of hydrogels (J. Mater. Chem. B. 2015, 3, 2286; ACS Appl. Mater. Inter. 2018, 10, 13304; Soft Matter. 2018, 14, 8401). We have improved the “Morphology of Hydrogels” section and added the references in the proper place in the revised manuscript (line 352-355).

44. Line 333: “before test” do you mean before SEM analysis?

Response: Thank the reviewer for the comment. We have improved the “Morphology of Hydrogels” section and changed “before test” to “before SEM observation” (line 354).

45. Line 336: better explain the composition of the solution you used.

Response: Thank the reviewer for the comment. We have added the composition of the buffer solution in the revised manuscript. The buffer solutions were adjusted using HAc and NaAc for pH 3 buffer solution, and Na2CO3 and NaHCO3 for pH 10 buffer solution, and the sodium chloride was used to adjust the ionic strength (lines 360-362).

46. Line 338: detail the considered time points.

Response: Thank the reviewer for the comment. We have added the time points in the revised manuscript (line 364).

47. Line 343: detail the considered time points.

Response: Thank the reviewer for the comment. We have added the time points in the revised manuscript (line 370).

48. Line 344: “maximum force” better explain what you mean as maximum force.

Response: Thank the reviewer for the comment. The “maximum force” is a parameter set in the program using DMA Q800 for compression test. It is referred to our previous studies and we have added the references in the proper place (line 374).

49. Line 347: you have to better explain the method used to load the hydrogels with triclosan; did you investigate the quantity of drug loaded in the hydrogels?

Response: Thank the reviewer for the comment. We forgot to add the preparation of triclosan preloaded hydrogels in the “Materials and Methods” section. In the revised manuscript, we have added the preparation process (lines 345-347). Besides, the drug entrapment efficiency (DEE) and drug loading efficiency (DLE) of the Gel-1, Gel-2, Gel-3 and Gel-4 were investigated (lines 193-195).

50. In vivo tests: you have to explain which is the rationale for that experiment.

Response: Thank the reviewer for the comment. We have added the references and more details (authorization) for the in vivo test in the revised manuscript.

51. Line 358: report the number of the authorization for the in vivo experiment

Response: Thank the reviewer for the comment. We have added the references and more details (authorization) for the in vivo test in the revised manuscript.

52. Line 366: you need to perform statistical analysis to check possible statistical differences among the different hydrogels.

Response: Thank the reviewer for the comment. We have added detailed statistical analysis results using Student’s t-test between each group. In this study, * indicates significant difference (p < 0.05); ** indicates significant difference compared with all other conditions (p < 0.01).

53. Lines 380-382: is it a refuse? please revise this part.

Response: Thanks for the reviewer’s comment. The words are from the given template and we have deleted them in the revised manuscript.

Reviewer 2 Report

Han etal., reported peptide-based bis-acrylate and acrylic acid hydrogels. The peptide. SEM and weight loss characterized swelling test conducted with pH and biodegradation of hybrid hydrogels. The author tested mechanical and cytotoxicity properties of the hydrogels.  Hydrogel formation for drug release and antibacterial also proved. Finally, investigators bis-acrylate/AAc hydrogel with stimuli-sensitivity and biodegradable property for wound dressing application.

Swelling Ratio of Hydrogels; should be in the linear graph formate; Here is an appropriate reference for similar hydrogel; pH-responsive MMT/Acrylamide Super Composite Hydrogel: characterization, Anticancer Drug Reservoir, and Controlled Release Property, Biochemistry and Biophysics (BAB) Volume 1 Issue 3, September 2013 

XRD, TGA and TEM analysis of hydrogels are missing 

The rationale behind the selection of S. aureus  for antibacterial activity test 

Materials and method sections writing is very poor and must be provided full details 

Author Response

Reviewer 2#

Han et al., reported peptide-based bis-acrylate and acrylic acid hydrogels. The peptide. SEM and weight loss characterized swelling test conducted with pH and biodegradation of hybrid hydrogels. The author tested mechanical and cytotoxicity properties of the hydrogels. Hydrogel formation for drug release and antibacterial also proved. Finally, investigators bis-acrylate/AAc hydrogel with stimuli-sensitivity and biodegradable property for wound dressing application.

Response summary: Thank the reviewer for the valuable comments. As the reviewer suggested, we have reorganized the Introduction section to provide an appropriate context and background of previous works of pH-responsive and biodegraded hydrogels. We have improved Figure 1&8, checked the statistical differences of hydrogels, and had more detailed discussion in the revised manuscript. Besides, we have added more information to the clearly explain Materials and Methods. Finally, the revised manuscript has been carefully checked to avoid any inappropriate terms, typos and other mistakes to improve the clarity and readability. We hope that the reviewer finds our responses satisfactory and convincing. The responses to comments are listed below:

Swelling Ratio of Hydrogels; should be in the linear graph formate; Here is an appropriate reference for similar hydrogel; pH-responsive MMT/Acrylamide Super Composite Hydrogel: characterization, Anticancer Drug Reservoir, and Controlled Release Property, Biochemistry and Biophysics (BAB) Volume 1 Issue 3, September 2013

Response: Thanks for the reviewer’s comment. We have redrawn the swelling ratio in the linear format (revised Figure 4a-c) and reset the y-axis using the same maximum values. Besides, as reviewer #1 suggested, we have added a figure (revised Figure 4d) to show the trend of swelling vs pH for each hydrogel.

Figure 4. Swelling ratio of Gel-1, Gel-2, Gel-3 and Gel-4 hydrogels in (a) pH = 3, (b) pH = 7 and (c) pH = 10. (d) The trend of swelling vs pH of each hydrogel.

XRD, TGA and TEM analysis of hydrogels are missing.

Response: Thanks for the reviewer’s comment. We have learnt the study the reviewer provided in which the researchers used XRD to determine the presence of montmorillonite with a strong peak, while in our study there was no comparison between these samples. Moreover, XRD is used to determine the atomic and molecular structure of crystals. In this study, the synthesized linear peptide had no strong diffraction peaks at 2θ value range of 0-40o so that we had not added the XRD test in the manuscript (Phys. Chem. Chem. Phys., 2015, 17, 6328).

Thermal stability: the hydrogels in this study show the enzymatic biodegradation property (Figure 5) and are thermos-stable which will had no weight loss in TGA test, thus we did not conduct the TGA test.

Morphology observation: SEM is the most common method used for the morphology observation of hydrogels, and the porous structure of the freeze-dried hydrogels can be investigated from SEM images (J. Control. Release, 2000, 69, 169; ACS Appl. Mater. Inter. 2012, 4, 2618; Biomaterials, 2009, 30, 6844). In this study, the morphology of hydrogels was observed by SEM as shown in Figure 3.

The rationale behind the selection of S. aureus for antibacterial activity test.

Response: Thanks for the reviewer’s comment. We have added the rationale and the relevant reference for the selection of S. aureus in antibacterial activity test (line 279-281).

Materials and method sections writing is very poor and must be provided full details

Response: Thank the reviewer for the comment. We have improved the “Materials and Methods” section to provide more details on the characterizations.

The main changes are listed below:

1. We have rewritten the characterization of hydrogel in a separate form to give more details of each test;

2. The preparation process of triclosan preloaded hydrogels was added in the “Preparation of Peptide Based Bis-acrylate/AAc Hydrogel” subsection;

3. The method of measuring the pore size of hydrogels was added in the “Morphology of Hydrogels” subsection;

4. The composition of the buffer solution with different pH values has been explained in “Swelling Ratio of Hydrogels” subsection;

5. The size of hydrogels for the compression test has been added in “Compressive Modulus of Hydrogels” subsection;

6. The details of drug release characterization have been added in “In vitro Drug Release from Hydrogels” subsection;

7. The details of biocompatibility test have been added in “Biocompatibility Evaluation” subsection;

8. Statistical analysis has been performed to check the statistical differences.

Round 2

Reviewer 1 Report

The Authors have revised the manuscript taking into consideration the reviewer's comments so that the quality of the work has been improved and the manuscript appears clearer

Reviewer 2 Report

I am agreed with the author's revision